# Forward Osmosis (FO) Membrane Fouling Mitigation during the Concentration of Cows’ Urine

**DOI:** 10.3390/membranes12020234

**Published:** 2022-02-18

**Authors:** Mokhtar Guizani, Ryusei Ito, Takato Matsuda

**Affiliations:** 1Environmental Engineering Division, Hokkaido University, Sapporo 060-8628, Japan; ryuusei.ito@gmail.com; 2Graduate School of Engineering, Hokkaido University, Sapporo 060-8628, Japan; takato.matsuda.ch@gmail.com

**Keywords:** forward osmosis, osmotic backwash, chemical cleaning, fouling, membrane

## Abstract

FO membrane fouling mitigation during the concentration of cows’ urine was investigated. In particular, the effects on the permeability recovery of cleaning methods such as membrane washing with deionized (DI) water, osmotic backwash, and chemical cleaning were studied. The characterization of foulants that accumulated on the membrane surface was found to be rich in sugars and proteins. The foulants were effectively removed by de-ionized water circulation (washing) and osmotic backwash. While osmotic back was more effective, it did not fully recover the permeability of the membrane. The foulants absorbed in the membrane pores were found to be mainly composed of sugars. Soaking the membrane in a solution of NaClO enabled the removal of foulants absorbed inside the membrane. It was revealed that soaking in 1% NaClO solution for 30 min achieved the best results (83% permeability recovery), while soaking for a longer time (10 h) using 0.2% NaClO resulted in counterproductive results.

## 1. Introduction

Despite the many fluctuations, global fertilizer demand has witnessed an annual increase since the 1970s and is expected to grow at an annual rate of 1.7%, as reported by the International Fertilizer Industry Association [1]. Furthermore, and according to the Food and Agricultural Organization (FAO) of the United Nations, the demand for fertilizer will continue to increase [2]. This increase in demand induces a rise in fertilizer prices, which constrains farming business and presents a threat to food security.

Interestingly, while nitrogen-based fertilizer production demands a large amount of energy, fertilizers such as phosphorus and potassium are non-renewable and unequally distributed across the world. Indeed, about 75% of phosphate reserves can be found in Morocco and the Western Sahara alone [3].

To meet the increasing demand for fertilizers and ensure local access to them, it is important to consider non-conventional sources. Livestock liquid waste streams are rich in plant nutrients such as nitrogen, phosphorus and potassium. According to Nevens W.B., the urine of dairy cows comprises one-third to one-half of nitrogen and three-fourths or more their excreta (urine and feces) is composed of potassium [4]. Harshbarger and Nevens report that from 12 to 16 pounds of nitrogen and 10 to 12 pounds of potassium may be found in urine for every ton of excrement [5]. While the direct application of livestock liquid waste streams is possible, this practice has several drawbacks, such as the decrease in the soil’s nitrogen-fixing capacity [6,7,8]. In addition, the proper balance of nutrients cannot easily be met, and the bulky volumes of these liquid streams increases the cost of storage and transportation, besides being a source of bad odors. In Japan, despite the decline in the number of livestock animals, approximately 8 million tons of urine from dairy cattle alone is generated per year, and the total amount of dairy farm liquid waste could be double that if we include dry cows and feeder cattle [9]. Current management practices for liquid waste streams in dairy farms in Japan include aerobic slurry treatment and land irrigation using treated or untreated liquid fractions to recycle nutrients [10]. The reported problems with current management systems include the huge cost required for storage tanks, as well as the smell and imbalance of nutrients [10].

Hence, a reduction in the volume of livestock liquid waste streams is necessary for successful nutrient recovery and, potentially, the production of fertilizers at proper nutrient ratios. Several volume reduction techniques can be employed. These are exemplified by evaporation, reverse osmosis (RO)-driven membrane filtration, freeze-drying, vacuum evaporation and electrodialysis, among others [11,12,13,14,15,16]. However, these concentration techniques demand a great deal of energy. Hence, forward osmosis (FO), a membrane-based process driven by the naturally occurring osmotic pressure difference (ΔΠ) between two solutions separated by a semi-permeable membrane, has emerged as an attractive method for the concentration of dairy farm liquid waste streams. Solutions of high concentration are called draw solutions (DS) while the low-concentration solution is called the feed solution (FS). The water flux passing through the membrane is proportional to the osmotic pressure difference, ΔΠ. Unlike the reverse osmosis process, the FO process is not pressure-driven, with the advantage of low energy consumption. It is reported that the implementation of FO for milk concentration could save 44% energy, 24% steam, 80% investments costs, and 50% operating costs [17]. Moreover, it is believed that the FO process has high resistance against membrane fouling in seawater desalination and wastewater treatment. Guizani et al. successfully concentrated wastewater using RO brine as a draw solution with no significant fouling [18]. Nikiema et al. applied the FO process to reduce the urine volume to 20% with 5 mol/L NaCl solution [19]. Nikiema’s study revealed that more than 90% of potassium and phosphate, and between 60% and 65% of nitrogen, were recovered. The increased recovery of nitrogen could be achieved by adjusting pH to reduce the diffusion of ammonia to draw the solution side. However, since cow urine contains a high concentration of organic matter and scaling precursors, the accumulation of organic matter and/or inorganic precipitations is likely to induce fouling and scaling in FO membranes. Indeed, in our previous study, the overall membrane permeability, which was calculated from water flux through the membrane and the osmotic pressure difference between cow urine and DS, decreased with repeated concentration processes, which suggests the occurrence of membrane fouling [20]. 

Although they are less prone to fouling, several researchers investigated FO membrane fouling during wastewater treatment [21,22,23,24,25,26]. It is well known that membrane fouling depends on feed stream characteristics, among other factors. However, cows’ urine and dairy farm liquid waste have not been extensively studied. To the best of our knowledge, no studies on FO fouling during the concentration of cow urine have been reported. Thus, this study investigated FO membrane fouling during repeated urine concentration cycles. A countless number of factors influence FO membrane fouling, including operational conditions (e.g., crossflow velocity and temperature), FS and DS characteristics (e.g., composition and concentration), membrane properties, and membrane orientation [26,27,28]. In this study, experiments were conducted at room temperature, using a 5 M NaCl draw solution, hydrolyzed cows’ urine as an FS, and the membranes’ active layer-facing feed solutions. A total of 5 M NaCl was used because Nikiema et al. found that a five-times concentration of urine cannot be achieved using a lower molarity [19]. The five-times concentration was dictated by the economic feasibility of recovering fertilizers from urine in comparison with commercial fertilizers [29]. Indeed, if concentrated fewer than five times, fertigation with urine cannot compete with commercial fertilizers in terms of storage and transportation costs, due to its large volume. We assumed that FO fouling during the concentration of cow urine would not be significant and that membrane performance would be easily recoverable. Hence, this study explores how fouling progresses, the major foulants and their characteristics. Then, it explores the effect of cleaning protocols to reduce FO membrane fouling during the concentration of cows’ urine.

## 2. Materials and Methods

### 2.1. Reagents, Feed, and Draw Solutions 

All the chemicals used in this study were of analytical reagent grade. Cow urine, collected from Hokkaido University experimental farm (Sapporo, Japan), was used as a feed solution after hydrolysis. The hydrolysis of cow urine was performed by the addition of urease into collected urine, and it was kept at room temperature for more than 1 day to complete the urea hydrolysis reaction. Draw solution (DS) was prepared by dissolving analytical-grade NaCl salt in de-ionized (DI) water.

### 2.2. Forward Osmosis (FO) Experimental Set-Up 

The schematic diagram of the lab-scale FO set-up is illustrated in Figure 1a. It consisted of two symmetric flow channels (Figure 1b) separated by a flat-sheet asymmetric cellulose tri-acetate CTA (Fluid Technology Solutions, Albany, OR, USA), and feed and draw solution tanks. The effective membrane area was 98 cm^2^. The membrane was made of a thin active layer (AL) deposited on a thick polyester screen support layer (SL). The operating pH of the membrane was in the range of from 2 to 11 according to the manufacturer’s datasheet. The membrane was soaked in DI water for at least one day before use. The active layer of the membrane was set in the feed solution side (AL-FS). The feed solution consisted of 400 mL of hydrolyzed cow urine, while 800 mL of 5 mol/L NaCl (Wako pure, Osaka, Japan, chemical grade) solution was used as the draw solution. Feed and draw solutions were circulated using peristaltic pumps. The solutions were circulated for 20 h in co-current mode at a crossflow of 14 L/hour (0.2 m/s) to limit the effect of concentration polarization. The volumes of feed and draw solutions were used to achieve a five-times concentration in 20 h. All the FO experiments were conducted at room temperature.

The cow urine concentration experiment was repeated for several cycles under the same conditions.

### 2.3. Permeability Coefficient Calculation 

The membrane’s performance can be described using its permeability coefficient. A drop in membrane permeability reflects a drop in membrane efficiency, which could be interpreted by fouling and scaling development. The permeability coefficient (P) expressed in [g/m^2^/s/kPa] is obtained from Equation (1) as a function of water flux (J_W_ in [g/m^2^/s]) passing through the membrane from the feed solution side to the draw solution and osmotic pressure difference (Δπ in kPa) between the feed solution and the draw solution. The permeability coefficient is the slope of the plot showing water transmission flux in a “y” axis versus osmotic pressure difference in the “x” axis.
P = Jw/Δπ,(1)

Water transmission flux through the membrane from the feed solution to the draw solution was experimentally determined by measuring the change in draw solution mass M(g). As per mass conservation laws, the transmission flux J_w_ of water was determined by differentiating the mass of diffusing atoms M(g) through a unit area A(m^2^) of the membrane used in the experiment per unit time t(s), as shown in Equation (2) [30].
J_w_ = A^−1^ × dM/dt,(2)

The osmotic pressure difference Δπ (KPa) was calculated using Equation (3), where R[kPa/K/mol] is the gas constant, T[K] temperature, and Σ a_(draw) and Σ a_(feed), respectively, are the sum of ion activities in the draw and feed solutions, respectively.
Δπ = RT (Σ a_(draw) − Σ a_(feed),(3)

### 2.4. Analytical Approach and Foulants Characterization

Cake layer composition analysis was performed using infrared spectroscopy to determine functional groups. The composition of the samples (2.5 mg each) from the cake layer were analyzed by Fourier transform infrared spectroscopy FTIR (FTIR-8400S, SHIMADZU, Kyoto, Japan). Precipitates from cow urine were obtained after cow urine filtration using a membrane filter (C045A090C, ADVANTEC, Tokyo, Japan). Cow urine precipitates were also analyzed in the same way for reference. Similarly, foulants contained inside the membrane pores were analyzed using FTIR. The active layer of the new and used membrane was first dissolved into dichloromethane (Wako pure drug, for high-speed liquid chromatography), and the liquid portion containing the dissolved CTA was transferred to the Falcon tube. The solvent was then evaporated by natural drying. All the samples were then freeze-dried (FDU-2100, EYELA, Tokyo, Japan) for 2 days under a 10 Pa vacuum and −86 °C. 

Moreover, 50 mg of the freeze-dried samples was used to measure T-N and T-C using the CN analyzer (SUMIGRAPH NC-220F, Sumika Chemical Analysis Center, Osaka, Japan). Simultaneously, hydrochloric acid (Wako pure agent, special grade) was added to the 50 mg sample and the same measurements were made. In addition, the inorganic elements (Na^+^, K^+^, Ca^2+^, Mg^2+^, Si, P, S) were measured using the optical emission spectroscopy method using inductively coupled plasma (ICP) (ICP-9000, SHIMADZU, Kyoto, Japan). All the measurements were performed three times, and the average value was calculated.

### 2.5. Physical Cleaning

To recover membrane permeability, physical cleaning was performed. In the physical cleaning experiment, two identical cow urine concentration experiments were conducted in parallel and operated in the same way. Experiment one (1) was stopped after 16 cycles and the membrane was disassembled for cake analysis. In experiment two (2), the operation continued for a total of 46 cycles, and then the membrane was disassembled for cake analysis. 

Physical cleaning was performed after the 5th, 10th, and 15th cycles for experiments 1 and 2. Additional physical cleaning for experiment two was conducted after the 25th, 30th, 35th, 40th, and 45th cycles.

During physical cleaning, except for cleanings conducted after the 15th, 30th, and 45th cycles, the feed and draw solutions were replaced with ultrapure water, and the membrane surface was physically cleaned by circulating water in the flow path for 2 h. This type of cleaning is referred to as cleaning by water circulation. The cleaning conducted after the 15th, 30th, and 45th cycles consisted of an osmotic backwash where FS and DS were replaced by NaCl (2M) and DI water, respectively. Subsequently, the concentration experiment was conducted and the change in the permeability coefficient was determined. 

### 2.6. Chemical Cleaning

Despite physical cleaning, the membrane’s permeation coefficient was not fully restored. Since the permeability coefficient was not fully restored by physical cleaning, it was expected that the cause of the deterioration in the permeability of the membrane was organic matter adsorbed inside membrane pores, which could not be removed by osmotic backwash. It should be noted that, according to the foulant characterization analysis, organic foulants were found in the cake layer. Hence, in this section, we describe how we attempted to recover the permeation coefficient by chemical cleaning. Cleaning was performed with sodium hypochlorite and sodium hydroxide. Sodium hypochlorite and alkali are effective at removing foulants because they strongly dissolve organic materials such as amide bonds, lipids, and polysaccharides associated with proteins [31]. On the other hand, as CTA membranes can be damaged under extreme pH values and chemical concentrations, the chemical cleaning conditions were chosen within such a range that sodium hypochlorite would not significantly affect the ion permeability of the CTA membranes [31]). Immersion wash was performed under conditions of NaOH (pH 10) and 1% NaClO (PH7, 10). Hydrochloric acid (Wako pure, special grade) and sodium hydroxide (Wako pure, special grade) were used to adjust the pH. Immersion was carried out for 30 min each time. 

To understand the change in cleaning effects due to differences in chemical concentrations and the soaking time, cleaning experiments were conducted by varying the chemical concentration and soaking time. Experimental conditions were set so that the CT value (concentration × time) remained constant. Thus, membranes were soaked for 1 h in 0.5% NaClO solution and for 2.5 h in 0.2% NaClO solution, respectively. 

Then, using the same solution (0.2% NaOCl), the soaking time was varied to 5 and 10 h, respectively.

The cleaning conditions are summarized in Table 1 below.

## 3. Results

### 3.1. Changes in the Permeability Coefficient

Membrane permeability was investigated during several cycles of repetitive cow urine concentration. Figure 2 shows the change in the permeability coefficient of the membrane in the repetitive concentration experiments, calculated from the initial flux and the difference in the osmotic pressure of each concentration cycle. The permeability coefficient showed a decreasing trend for each new cycle. Despite regular cleaning attempts, the permeability dropped to less than 20% of the permeability of a virgin membrane after 25 cycles. These observations are consistent with previous experiments conducted by the same researchers [20]. It is worth mentioning that, after the first cycle, the drop in permeability was relatively rapid (from 100% to nearly 60%); then, the permeability dropped gradually. Similar observations were confirmed after each cleaning. Further discussions are presented later in the paper regarding the effect of cleaning on permeability recovery.

Knowing that experimental conditions were kept the same for all concentration cycles, the observation indicates that the drop in membrane permeability was due to FO membrane fouling. Similar observations of flux drop, indicative of fouling formation, were reported by X. Liu et al. [22]. Moreover, the formation of a thick dark brown cake layer on the membrane surface was observed (Figure 3). This cake layer is indicative of membrane fouling and explains the drop in membrane permeability. 

It is also important to mention that, in early concentration cycles, 20 h was sufficient to concentrate urine five times, while a five-times concentration could not be achieved as fouling developed and the membrane flux dropped. 

### 3.2. Fouling Characteristics

Figure 3 shows the membrane conditions after several concentration cycles. From these images, it can be seen that severe dark brown cake layers formed on membrane channels. In Figure 3a, some of the foulants were washed away following physical cleaning. The composition of foulants was analyzed using FTIR. The IR spectra illustrated in Figure 4 indicate the presence of sugars in the cake layer, as observed in the I-H bond peaks near 3400 cm^−1^, and C-O-C bond peaks near 1100 cm^−1^. In addition, the presence of C=O and C-N and N-H bonds (19) in 1560 cm^−1^ and 1400 cm^−1^ indicates the presence of amide bonds contained in the protein. Hence, it could be confirmed that the materials that make up the organic matter present in the cake layer on the membrane surface contain sugars and proteins. The peak height of the C=O bond for the 46-cycle experiment was higher, suggesting that it contains more protein or carbonates than 16-cycle experiments. The urine precipitates in the hydrolyzed urine showed a similar spectrum to cake layers, indicating that these precipitations formed the cake layers on the membrane after urine concentration. The element analysis showed that C, N, and O (organic matter) were major components in cake layers. As illustrated in Figure 5, the 46-cycle experiment showed a higher concentration of Ca and Mg in the foulants than their concentration after the 16-cycle experiment.

### 3.3. Effect of Physical Cleaning

Figure 6 illustrates the normalized permeability coefficient, calculated from the water flux and osmotic pressure difference in the early stage of operations in each cycle, for Experiment 1 and Experiment 2. The normalized value was obtained by dividing the permeability coefficient at the beginning of each cycle by the permeability coefficient at the beginning of the initial concentration cycle. For the sake of simplicity, error bars are not shown; however, it was confirmed that the standard deviation was insignificant.

Graphs clearly show that permeability could be recovered after each cleaning. After the physical cleaning of the membrane surface by the DI water circulation, an average recovery effect of 69% permeability was achieved, compared with the permeability after the previous wash. It should be noted that, after physical cleaning (both in the case of simple water recirculation and osmotic backwash), the permeability dropped sharply after the first cycle, and then continued to drop at a slower rate. This can be explained by the fact that, with an early operation and given the membrane’s roughness, foulants easily stick to the membrane pores and rough surfaces, contributing to a quick drop in permeability. Once pores and rough surfaces are clogged, the fouling progresses slowly. Further studies are needed to confirm this assumption, using SEM analysis or a similar technique.

Cleaning with an osmotic backwash shows higher permeability recovery in comparison with simple washing with water circulation. A recovery of 84% of the virgin membrane permeability was shown. This finding suggests that osmotic backwash removes the foulants absorbed inside membrane pores that cannot be removed by simple water circulation, in addition to the surface foulants that are easy to remove via water circulation. The non-full recovery of permeability by osmotic backwash could be explained by the fact that the foulant that accumulates inside membrane pores cannot be completely washed out. It was also confirmed that some of the foulants that adhered to the membrane surface and were not removed by DI water circulation were peeling off after osmotic backwash was applied. In brief, although increasing the frequency of physical washing by water washing alone has some effect on recovering the permeability coefficient, the removal effect of membrane surface fouling is limited, and cleaning using osmotic backwash effectively restores the permeability drop caused by fouling on the membrane surface. This shows the possibility of continuous urine concentration by restoring the permeability coefficient with physical cleaning. However, it should be noted that the full recovery of permeability by physical cleaning (DI water circulation and osmotic backwash) could not be achieved. Regarding the foulant characterization results, foulants are mainly of an organic nature (sugars and proteins). Hence, it is believed that chemical cleaning using sodium hypochlorite and sodium hydroxide could help recover the permeability. 

### 3.4. Chemical Cleaning

The results of chemical cleaning using NaOH and NaClO are shown in Figure 7. Here, it should be noted that chemical washing was performed on a fouled membrane where no physical cleaning was performed. Cleaning with NaOH (pH 10) and 1% NaClO (pH 7) showed little recovery of the permeability coefficient, while washing with 1% NaClO (pH 10) showed a higher permeability recovery effect of 83% for the new membrane. Figure 8 shows the status of the cake layer before and after chemical cleaning, indicating a good cleaning effect. 

### 3.5. Chemical Cleaning: Effect of Concentration and Soaking Time

As illustrated in Figure 9a, the membrane immersed in 0.5% NaClO for 1 h and the membrane immersed in 0.2% NaClO for 2.5 h both showed similar permeability recovery (69%). The same figure shows that soaking the membrane in 1% NaClO solution for 30 min achieved higher permeability recovery (83%). However, soaking the membrane in 0.2% NaClO for a longer time did not achieve higher permeability recovery. Indeed, for a soaking time of 5 h, an improvement in permeability recovery was observed (76%) in comparison with 2.5 h of soaking time using the same solution (0.2% NaOCl), but this value remained lower than the recovery rate achieved when soaking in 1% NaOCl for 30 min. Interestingly, an increased soaking time resulted in a drop in permeability recovery (59% only). The possible reason for this unexpected drop in permeability recovery, despite the extended soaking time, is the possibility that NaClO and CTA reacted due to the excessive contact time, resulting in a change in the potential of the membrane surface [32]. 

On the other hand, since the permeability coefficient after chemical treatment with 1% NaClO was the highest, simple CT values alone do not determine the cleaning effect. Further investigation is needed to understand how chemical cleaning works.

### 3.6. Characterization of Foulants Absorbed Inside Membrane Pores

The structure of the foulants absorbed inside the membrane pores, which could not be removed by physical cleaning, was analyzed by FTIR. The analysis was performed on a new membrane, a membrane operated for 16 cycles and a membrane operated for 30 cycles. The membranes used were analyzed right after the physical cleaning and before chemical cleaning.

Figure 10 shows the results of the FTIR measurements. Peaks near 3400 cm^−1^ represent O–H bonds, peaks near 1750 cm^−1^ represent C=O bonds, and peaks near 1240 cm^−1^ and 1050 cm^−1^ represent C-O-C bonds, which are characteristic of CTA membranes. (20) The ratio of O-H bonds near 3400 cm^−1^, C–H bonds near 2900 cm^−1^, and C-O-C bonds near 1050 cm^−1^ increased in the membranes where cow urine concentration was performed. Membranes operated for a longer period show a higher concentration. These structures indicate the presence of sugars. As shown in Figure 4, foulants on the surface of the membrane contain proteins, but as there are no protein peaks in foulants absorbed into pores inside the membrane that represents amide bonds, it is concluded that the proteins are present on the surface of the membrane as deposits that cannot be adsorbed inside the membrane and that can be removed by physical cleaning. Indeed, the proteins are larger than the membrane pores. Therefore, the organic matter in the foulant inside the membrane mainly comprises sugars. The reason the membrane’s permeability coefficient was restored by cleaning with NaClO is because these sugars were dissolved by NaClO.

## 4. Discussion

It was assumed that the FO membrane is less prone to fouling. However, in the case of cow urine concentrations, it was revealed that FO permeability dropped significantly due to fouling (Figure 2). Similar studies on FO fouling in challenging wastes such as distillery wastewater, olive oil mill wastewater, dairy whey dewatering, and juice concentration revealed that membrane fouling occurred [33,34,35,36]. Although it is not easy to compare the different studies due to variations in experimental set-ups, operation conditions and data interpretation, it was evident that the drop in membrane permeability in the case of cow urine was the most significant, with an 80% drop in permeability. 

Fouling in FO membrane is classified into internal and external fouling. While external fouling is present independently of membrane orientation, internal fouling presence depends on membrane orientation. When the active layer faces feed solution, the membrane is said to be in FO mode. When the support layer faces feed solution the membrane is said to be in pressure retarded osmosis mode (PRO). In PRO mode, the fouling mechanism is more complex than in FO mode. Indeed, in FO mode, an external cake layer is formed on the active layer. However, in PRO mode, small foulants can get inside support layer pores inducing an internal fouling in addition to the external cake layer. While physical cleaning can remove the external cake layer, internal fouling is hard to clean up and often requires chemically enhanced washing [37]. In our study, the membrane was used in FO mode, where the active layer faced cows’ urine. Our findings indicated 85% flux recovery using physical cleaning, which agreed with previous studies on flux recovery of fouled FO membranes [38]. In PRO mode, physical cleaning achieved 30% flux recovery only, indicating the dominance of irreversible fouling in PRO mode [38].

Moreover, our findings suggested that internal fouling has occurred during cows’ urine concentration experiment, indicating that nano-scale foulants clog the nano-pores of the active layer. This assumption agreed with our observation of diffusivity of molecules smaller than 200 KDa through FO membrane (unpublished). During their passage these molecules interact with the hydrophilic CTA membrane (contact angle of active layer 70 degree; negative zeta potential: −0.34 mV) and slowly clog the pores. An in-depth analysis of membrane properties, and foulants would provide us more detailed information on fouling mechanisms of FO membrane during cow’s urine concentration. 

It is worth recalling. that the experiment was conducted using a 5 M NaCl solution, resulting in a relatively high flux (>50 LMH). However, it is well known that beyond a critical flux, fouling is observed, and below the critical flux, flux decline with time does not occur [39]. Moreover, below the critical flux, cross-flow filtration helps overcome the effects of concentration polarization (CP). Beyond the critical flux, not only concentration polarization persists, but also foulants are pushed against the surface of the membrane under the high flux effects, which in return enhance the effects of concentration polarization [40]. This is in agreement with the observation of Wang et al, who observed that the surface coverage of membrane by a latex model foulant increased significantly beyond a critical flux of 28 LMH [40]. Hence, this could explain why severe fouling has occurred during cows’ urine concentration.

An increase in cross-flow velocity is recommended to reduce the effect of critical flux, which ultimately could result in high energy consumption. A second option to reduce the effect of critical flux is to perform the cows’ urine concentration into two or more steps using the relatively small osmotic pressure difference between the feed and draw solution.

Back to fouling mitigation, unlike pressure-driven membranes, FO fouling can be easily reduced through physical (DI water circulation and backwash) and chemical cleanings. A membrane performance restoration of more than 85% could be achieved using physical cleaning (rinsing with deionized water and osmotic back wash). According to Anita and Andre, higher membrane performance restoration could be achieved using physical cleaning during the treatment of olive oil mill wastewater (95%) and dairy whey dewatering (90%) [41]. 

Unlike the aforementioned cases, where more than 90% of membrane performance was regenerated by physical cleaning, this study revealed that chemical cleaning is required to remove foulants that are not removed by physical cleaning. In this study, chemical cleaning showed that 83% of membrane permeability could be recovered after soaking the membrane in 1% NaClO solution for 30 min. It should be noted that no reverse salt diffusion was observed following chemical cleaning, indicating that chemical cleaning did not cause damage to the membrane. In the work conducted by Liu et al., it was reported that chemical cleaning was more efficient than physical cleaning [22]. 

The results of chemical cleaning reported in this paper are for a fouled membrane where no physical cleaning was performed. Thus, a combination of physical and chemical cleaning is believed to be more effective in maintaining the flux through the FO membrane during the concentration of cows’ urine. Further studies are required to investigate the combination of physical and chemical cleaning.

Furthermore, analysis of the foulants showed that the foulants that attached to the surface of the membrane included organic materials containing sugars and proteins, as well as inorganic materials such as Ca and Mg, while the foulant absorbed inside the membrane was mainly composed of sugars (organic matter). However, across the 45 cycles, the mass percentage of inorganics (Ca and Mg) increased compared with their percentage after only 16 cycles. In this paper, we did not address the inorganic foulants (scale), as they seem to have no significant impact on permeability. However, in long-term operations, scale formation (due to Ca and Mg precipitation) is likely to be more significant than Ca and Mg accumulate over time. Further studies on scale formation and removal during FO operation are needed.

## 5. Conclusions

In this study, FO membrane fouling during the concentration of cow urine using the FO system was investigated. Repeated cow urine concentration reduced the membrane permeability to less than 20% of the permeability of the virgin membrane and confirmed the development of severe membrane fouling. Analysis of the foulants attached to the surface of the membrane showed that they were mainly composed of organic materials containing sugars and proteins and some inorganic materials such as Ca and Mg. However, the foulants found inside membrane pores only comprise sugars. The physical cleaning of the membrane by circulating DI water in the flow path was able to achieve a 69% recovery of membrane permeability. However, some foulants attached to the surface of the membrane and inside the membrane pores could not be removed by simple washing methods. Hence, osmotic backwash was used. The osmotic backwash showed better cleaning effects, and better permeability recovery was achieved, reaching up to 84% of the permeability of a virgin membrane. However, the osmotic backwash was not able to fully recover membrane permeability. Thus, chemical cleaning was used. Chemical cleaning investigation revealed that soaking the membrane in 1% NaClO solution for 30 min recovered 83% of the permeability of a virgin membrane. This suggests that a good combination between physical and chemical cleaning would achieve a complete recovery of membrane performance. Interestingly, it was found that an extended period of soaking for a longer time (10 h) using 0.2% NaClO could have counter-productive results. 

## Figures and Tables

**Figure 1 membranes-12-00234-f001:**
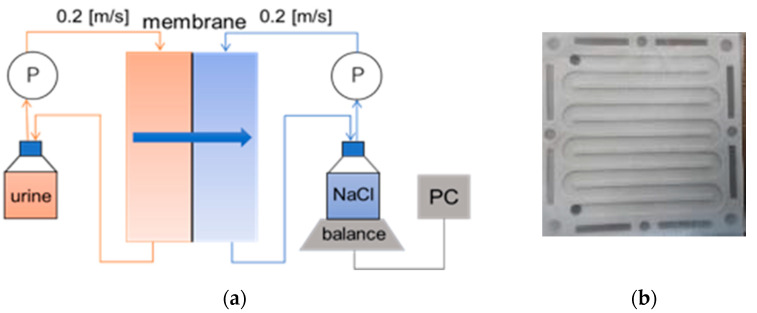
The schematic diagram of the urine concentration using the forward osmosis (FO) system: (**a**) experiment set-up, (**b**) membrane channels.

**Figure 2 membranes-12-00234-f002:**
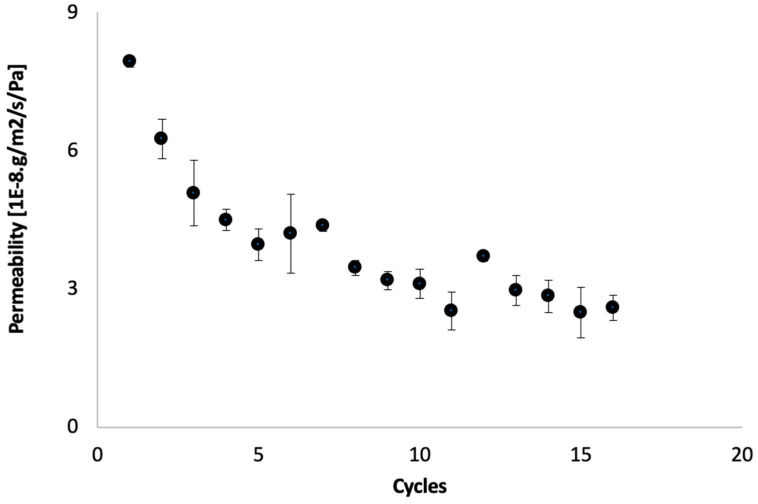
Membrane permeability at the start of each new concentration cycle (measured in steady flow conditions approximately 5 min after concentration cycle starts).

**Figure 3 membranes-12-00234-f003:**
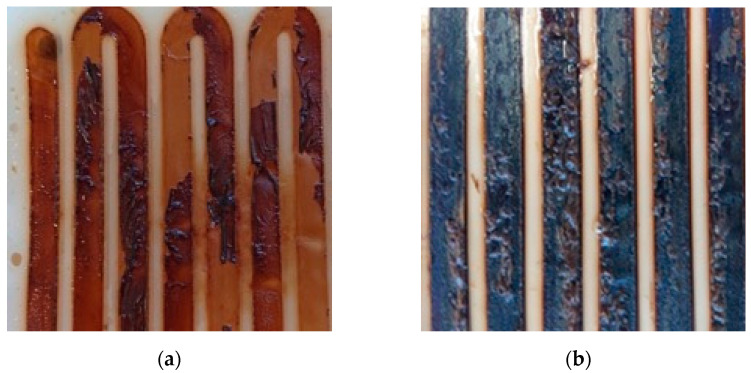
Fouling cake layer on the membrane channels after concentration experiments: (**a**) after 16 cycles, followed by osmotic backwash; (**b**) after 46 cycles. A caption on a single line should be centered.

**Figure 4 membranes-12-00234-f004:**
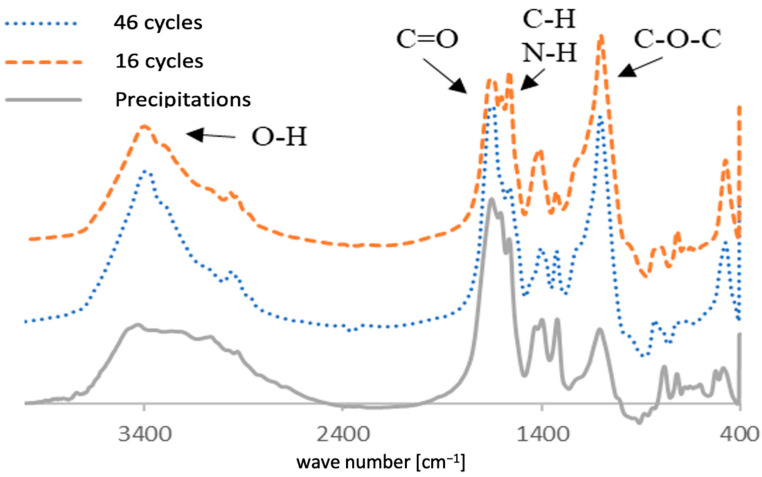
Fourier transform infrared (FTIR) characterization of foulants that accumulated on the surface of membrane.

**Figure 5 membranes-12-00234-f005:**
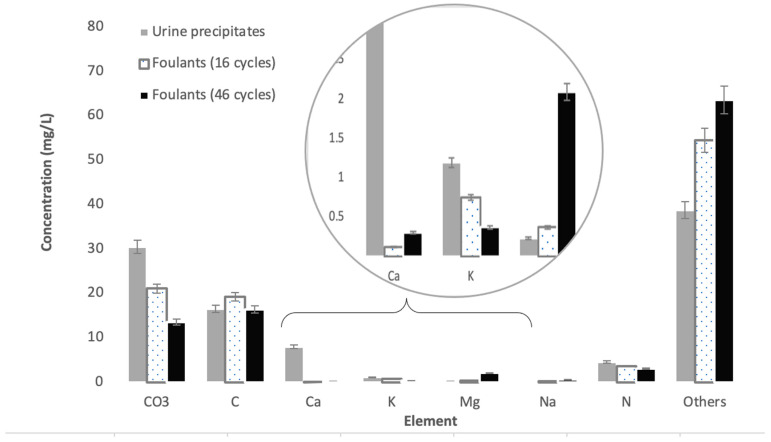
Chemical composition of foulants and urine precipitates; error bars show standard deviation.

**Figure 6 membranes-12-00234-f006:**
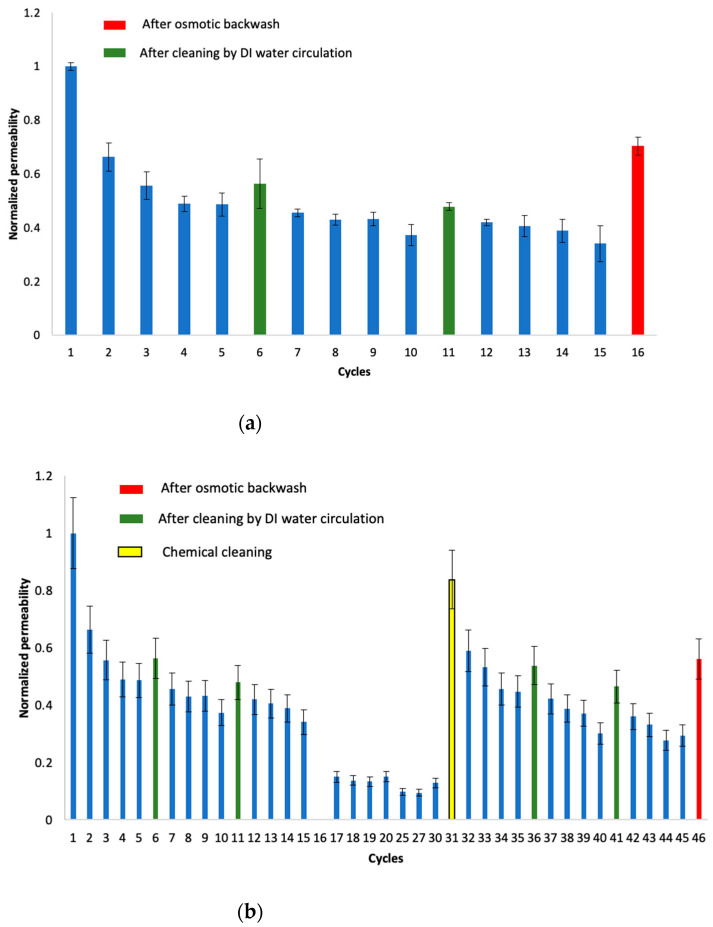
Effect of physical cleaning on permeability; (**a**) Experiment 1; (**b**) Experiment 2: the normalized value is obtained by dividing permeability coefficient at the beginning of each cycle by the permeability coefficient at the beginning of initial concentration cycle; error bars show standard deviation.

**Figure 7 membranes-12-00234-f007:**
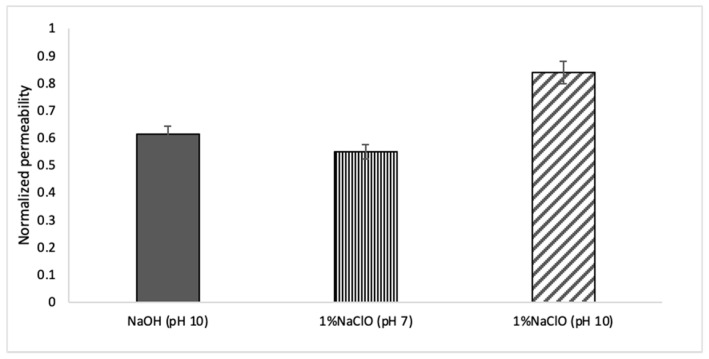
Effect of chemical cleaning on membrane permeability, the normalized value is obtained by dividing permeability coefficient at the beginning of each cycle by the permeability coefficient at the beginning of the initial concentration cycle; error bars show standard deviation.

**Figure 8 membranes-12-00234-f008:**
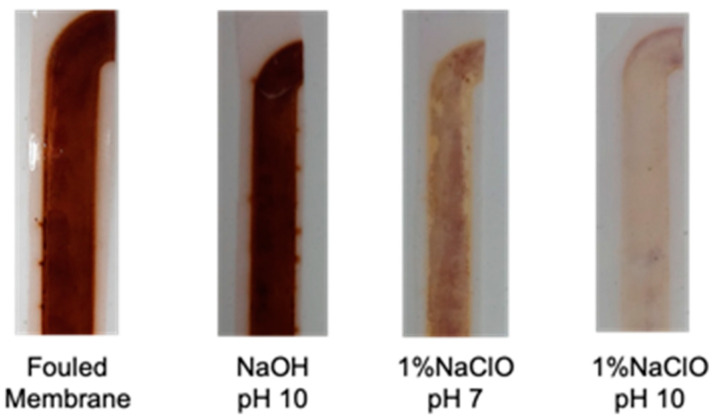
Visual observation of chemical cleaning effect.

**Figure 9 membranes-12-00234-f009:**
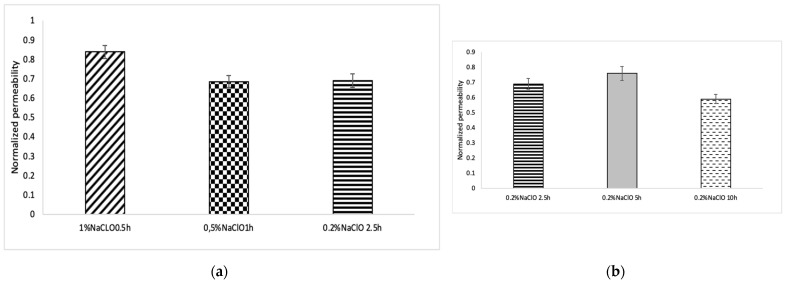
Effect of soaking time on membrane permeability recovery: (**a**) Effect of concentration and soaking time change at a constant concentration Time CT value (**b**)Effect of soaking time change at a fixed concentration; the normalized value is obtained by dividing the permeability coefficient at the beginning of each cycle by the permeability coefficient at the beginning of the initial concentration cycle; error bars show standard deviation.

**Figure 10 membranes-12-00234-f010:**
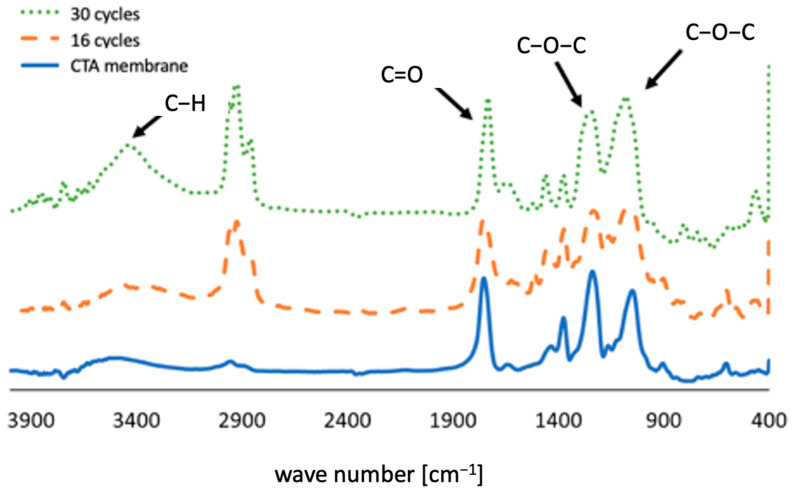
FTIR characterization of foulants absorbed inside membrane pores.

**Table 1 membranes-12-00234-t001:** Experimental conditions for chemical cleaning of fouled membranes.

	Chemicals	Soaking Time
Test 1	NaOH (pH 10) 1% NaClO (pH 7, 10)	30 min
Test 2	0.5 % NaClO (pH 10)	1 h
Test 3	0.2% NaClO (pH 10)	2.5, 5, 10 h

## Data Availability

Not applicable.

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
