# Peer review of "Forward Osmosis (FO) Membrane Fouling Mitigation during the Concentration of Cows’ Urine"

_membranes, 2022, doi:10.3390/membranes12020234_

Round 1

Reviewer 1 Report

Overall comment:

The manuscript needs to be edited and the writing style needs to be improved by a professional editor.

This work looks more like a technical report than a scientific publication. Lasks significantly from in-depth investigation and discussion. 

Abstract:

  • The  following two first sentences need to be removed as the reader of this journal are aware of membrane fouling issues (in general):

" Membrane fouling is a frequent problem for membrane-driven water and wastewater treatment. The forward osmosis (FO) membrane has emerged as an attractive non-fouling prone membrane. However, in solutions with high organics and scaling precursors, FO fouling and scaling are inevitable"

2. Materials and Methods

  • Figure 1.  does not show anything and adds no useful info. iT needs to be removed. 
  • Have you done any replicate? If not, please provide the reasons.

3. Results 

  • Figure 3 has no error bar and an incomplete legend.
  • The caption of Fig 5 is wrong. 
  • How did you normalize your results? It should be clearly stated for each figure. 
  • Fig. 6 is not clear, expressing the results in percentage does not always present accurate results. consider changing the presentation of this figure and add more explanation in the text. 

4. Discussion

This section is extremely short needs significant improvement.

6. Conclusion 

  • This section repeats the abstract. needs to be rewritten to show a more in-depth conclusion

Author Response

COVER LETTER

Date: January 23, 2022

Dear Editor & Reviewers

Enclosed is the revised and language proof edited version of the manuscript ID membranes-1578872. by Guizani M. et al. titled “Forward Osmosis (FO) membrane fouling mitigation during cows’ urine concentration”, which is being submitted for possible publication in the Journal of membranes. Authors would like to express their gratitude to the reviewers for their valuable comments and suggestions and their time spent to improve this work, as well as to the editor for considering the manuscript for possible publication in journal of Membranes. Care has been taken to improve the manuscript and address all reviewers’ concerns one by one. Following is the list of reviewers’ comments and queries followed by our responses to these comments and queries.

REVIEWER 1

Overall comment:

The manuscript needs to be edited and the writing style needs to be improved by a professional editor.

This work looks more like a technical report than a scientific publication. Laks significantly from in-depth investigation and discussion. 

 Answer:

Thank you for your valuable comment. However, we think that the understanding and characterization of foulants and finding the best permeate recovery protocols during cows’ urine concentration by FO system is of paramount importance both from technical point of view and scientific point of view. In addition, the changes we incorporated in the revised file during the revision (expansion of the Introduction, discussion section expansion and improvement.) as well as the response to the various comments raised by both reviewers, put more highlights on the scientific value of this work.

Kindly refer to the file with track changes

Abstract:

  • The following two first sentences need to be removed as the reader of this journal are aware of membrane fouling issues (in general):

" Membrane fouling is a frequent problem for membrane-driven water and wastewater treatment. The forward osmosis (FO) membrane has emerged as an attractive non-fouling prone membrane. However, in solutions with high organics and scaling precursors, FO fouling and scaling are inevitable"

Answer:  These sentences were deleted (kindly refer to track changes Page 1 Line 9).

  1. Materials and Methods
  • Figure 1.  does not show anything and adds no useful info. iT needs to be removed.

Answer: Figure 1 was deleted And Figures numbering was revised in manuscript.

  • Have you done any replicate? If not, please provide the reasons.

Answer: Yes, experiments were in triplicates. Figure 2 (in the revised version) was updated to show permeability coefficient instead of normalized values and error bars are shown in the figure. We have confirmed that standard deviation is insignificant, so in the rest of figures and for the sake of simplicity and clarity, we choose not to show error bars. 

  1. Results 
  • Figure 3 has no error bar and an incomplete legend.

Answer: All figures were renumbered, and revised. Figure 3 (before the revision), now denoted Figure 2 includes error bars and the legend has been updated. (kindly, refer to file with track changes Page 6 Line 610)

  • The caption of Fig 5 is wrong.

Answer: Same as above, Figure 5 (figure 4 in the revised file) was revised and caption corrected (Page 7 Line 756)  

  • How did you normalize your results? It should be clearly stated for each figure. 

Answer: “The normalized value is obtained by dividing permeability coefficient at the beginning of each cycle by the permeability coefficient at the beginning of initial concentration cycle.” This sentence was introduced in the manuscript in Page 8 Line.770 It is also inserted in figure showing normalized data (Figures 6, 7 and 9).

  • Fig. 6 is not clear, expressing the results in percentage does not always present accurate results. consider changing the presentation of this figure and add more explanation in the text. 
  1. Discussion
  • This section is extremely short needs significant improvement.

Answer: Thank you for this comment. We agree that discussion section was too limited. Discussion was improved. Kindly refer to Page 12 Lines 958 t0 Page 13 1026.

  1. Conclusion 
  • This section repeats the abstract. needs to be rewritten to show a more in-depth conclusion

Answer: Thank you for this comment. The conclusion was rewritten (kindly refer to the revised manuscript Page 13 Line 1052) .

Reviewer 2 Report

This manuscript characterizes the foulants and explores the effect of cleaning protocols to reduce forward osmosis membrane fouling during cows’ urine concentration.

1) Authors claim that "To our knowledge, no studies have been reported on FO fouling during cows’ urine concentration". This is the application aspect. What is the scientific contribution/novelty of this manuscript? Otherwise, this manuscript doesn't show transformative change approach.

2) What is the hypothesis?

3) Equations written in the "2.3. Permeability coefficient calculation" section need reference(s).

4) Why did you choose 5M NaCl draw solution for this study?

5) Conclusion section is similar to the abstract section. The major finding should be written in the conclusion section.

6) How do you apply the results of this study to industrial-scale?

Author Response

COVER LETTER

Date: January 23, 2022

Dear Editor & Reviewers

Enclosed is the revised and language proof edited version of the manuscript ID membranes-1578872. by Guizani M. et al. titled “Forward Osmosis (FO) membrane fouling mitigation during cows’ urine concentration”, which is being submitted for possible publication in the Journal of membranes. Authors would like to express their gratitude to the reviewers for their valuable comments and suggestions and their time spent to improve this work, as well as to the editor for considering the manuscript for possible publication in journal of Membranes. Care has been taken to improve the manuscript and address all reviewers’ concerns one by one. Following is the list of reviewers’ comments and queries followed by our responses to these comments and queries.

REVIEWER 2

This manuscript characterizes the foulants and explores the effect of cleaning protocols to reduce forward osmosis membrane fouling during cows’ urine concentration.

1) Authors claim that "To our knowledge, no studies have been reported on FO fouling during cows’ urine concentration". This is the application aspect. What is the scientific contribution/novelty of this manuscript? Otherwise, this manuscript doesn't show transformative change approach.

Answer: We would like to thank the reviewer for this question. We believe that, since no former studies have been reported on the use of forward osmosis to concentrate cows’ urine, there are still many scientific questions to be answered. Among this question, is how fouling progresses during cow’s urine concentration using FO? What kind of foulants are there? And is membrane’s permeability recoverable? These scientific questions were incorporated in the introduction to reflect the scientific importance of this work (Page 2 L144-151)

The paper answers these questions and revealed that severe fouling and drop in membrane’s permeability was observed. Physical cleaning and chemical cleanings together could regenerate the membrane performance.

2) What is the hypothesis?

Answer: It was assumed that FO membrane is less prone to fouling, (revision Page 2 L 149) but this study that this assumption is not valid. In case of cows’ urine significant fouling was observed. Unlike, most other application, physical cleaning was found not to be sufficient to fully recover permeability of the membrane. Rather, chemical cleaning is required. Hence, to use FO for cows’ urine concentration, regular physical cleaning and chemical cleanings are required.

3) Equations written in the "2.3. Permeability coefficient calculation" section need reference(s).

Answer; This equation is derived from mass conservation law, rate of mass change per unit area per time. We modified the sentences in the manuscript as follows: Reference was also added. The text is inserted in the manuscript in Page 4 L457 and it is as follows:

As per mass conservation law, the transmission flux Jw of water is determined by differentiating the mass diffusing atoms M(g) through a unit area A(m2) of the membrane used in the experiment per unit time t(s) as shown in equation (2) [30].

4) Why did you choose 5M NaCl draw solution for this study?

Answer: We update the manuscript by providing the explanation for the reason of using 5M NaCl as a draw solution in Page2 Line 144

The 5M. NaCl was used as it was found by Nikiema et al. that five times concentration of urine is hardly achieved using lower molarity [16]. The five-times concentration is dictated by the economic feasibility of fertilizers recovery from urine in comparison with commercial fertilizers [26]. Indeed, if concentrated less than 5 times, fertigation with urine cannot compete with commercial fertilizers, in term of cost due to large volume.

5) Conclusion section is similar to the abstract section. The major finding should be written in the conclusion section.

Answer: Thank you for this comment. The conclusion was rewritten (kindly refer to the revised manuscript Page 13 Line 1031)

6) How do you apply the results of this study to industrial-scale?

Answer: Thank you very much for this important question. Finding of this research suggests that if cows’ urine is to be concentrated using forward osmosis, fouling will be significant (drop of 80% of membrane permeability). However proper physical and chemical cleaning, would sustain the operation of the concentration process. It was assumed that physical cleaning would suffice, but this study reveals that physical cleaning alone would not be enough.

This sort of practical advice is included in the discussion section.

Sincerely yours

Guizani Mokhtar

Round 2

Reviewer 1 Report

I see some improvements in the revised version; however, the manuscript needs to be edited by a professional English editor.

I do not see my comments regarding the error bars for experimental data was fully respected esp from Fig 5 onward.

The discussion still needs significant improvement to sound like a scientific paper and not a technical report.

I would like to see the revised version esp the revised discussion.

Author Response

Dear Reviewer;

Dear Reviewer;

Authors would like to thank reviewers for their valuable comments. The manuscript has been revised following your reviewers comments. Special focus were given to the discussion section, errors bars inclusion and professional language proof editing. All the revisions can be tracked on the revised manuscript with track changes. Response to reviewer 2 comments can also be found the the attached file including response to reviewer 1.

Yours Truly 

Guizani Mokhtar

Authors would like to thank the reviewer for his valuable comments. Following, are our responses to the comments addressed one by one.

Your Truly 

Guizani Mokhtar 

Reviewer 2 Report

N/A

Author Response

Dear Reviewer;

Authors would like to thank reviewers for their valuable comments. The manuscript has been revised following your reviewers comments. Special focus were given to the discussion section, errors bars inclusion and professional language proof editing. All the revisions can be tracked on the revised manuscript with track changes. There is no special comments from reviewer 2 to be addressed.

Yours Truly 

Guizani Mokhtar